# Short Paper Submissions for the MICCAI KiTS21 Challenge

Y. J. Wu[1], Y.L. Zhao[1]

The aladdin5 team

7766963@qq.com

**Abstract.** We use nnUNet and also tried several methods to process and train the data.

**Keywords:** nnUNet, ROI

## 1 Introduction

Kidney cyst and tumor are very small compared to the whole volume of CT data, and so it's difficult to achieve ideal precision for normal training and prediction. We propose a method to train and learn the data efficiently.

## 2 Methods

We use nnUNet and also tried several methods to process and train the data.

### 2.1 Training and Validation Data

Our submission made use of the official KiTS21 training set alone.

### 2.2 Preprocessing

We use nnUNet preprocessing and also:

1. We extract the kidney ROI region using the bounding box of the left and right kidney;
2. We separate the left kidney, left kidney cyst, left kidney tumor, right kidney, right kidney cyst, right kidney tumor;

### 2.3 Proposed Method

Our method is as follows:

1. We use nnUNet.

2. We train the left kidney and right kidney together and then use a postprocessing script to split the left and right kidney. We also tried to train the left and right kidney separately, but found that the model always misinterpreted the left and right kidney for many cases. We use two networks to train the kidney, the first is nnUNet lowres mode, and the second only uses the ROI of kidney bounding box and do fullres training.

3. The we train the tumor and cyst of the left and right kidney separately, which means we have 4 more networks – left kidney tumor, right kidney tumor, left kidney cyst, and right kidney cyst. We only use the ROI of kidney bounding box and do fullres training, and add the left/right kidney as an extra input channel.

4. Finally, we combine all the separate results together, and most importantly, to put the ROI back to the full volume.

# 3 Results

We split a small set of the original data as the test set. Here are the dice results of the test set.

**Model1-kidney-lowres:**

|     | dice  |
| --- | ----- |
| std | 0.051 |
| max | 0.985 |
| min | 0.775 |
| med | 0.975 |
| avg | 0.960 |

**Model2-kidney-fullres:**

|     | dice  |
| --- | ----- |
| std | 0.028 |
| max | 0.993 |
| min | 0.813 |
| med | 0.979 |
| avg | 0.975 |

**Model3-left-kidney-tumor-fullres (16 test cases):**

|     | dice  |
| --- | ----- |
| std | 0.138 |
| max | 0.978 |
| min | 0.368 |
| med | 0.926 |
| avg | 0.883 |

**Model4-right-kidney-tumor-fullres (15 test cases):**

|     | dice  |
| --- | ----- |
| std | 0.290 |
| max | 0.976 |
| min | 0.0   |
| med | 0.872 |
| avg | 0.742 |

**Model5-left-kidney-cyst-fullres (12 test cases):**

|     | dice  |
| --- | ----- |
| std | 0.339 |
| max | 0.904 |
| min | 0.0   |
| med | 0.727 |
| avg | 0.579 |

**Model6-right-kidney-cyst-fullres (11 test cases):**

|      | dice  |
|------|-------|
| std  | 0.305 |
| max  | 0.956 |
| min  | 0.014 |
| med  | 0.888 |
| avg  | 0.711 |

## 3  Discussion and Conclusion

We didn't use any new algorithms and just used the nnUNet to play with the data. Hopefully we could add some improvements based on our current work for the next step.

## Acknowledgements

Thanks to the nnUNet.

## References

1.      Smith, T.F., Waterman, M.S.: Identification of Common Molecular Subsequences. J. Mol. Biol. 147, 195--197 (1981)