# OpenReview forum: "kits21_short_paper_aladdin5_team"
_MICCAI.org/2021/Challenge/KiTS — Submitted to KiTS21 Challenge_

### Official Review · Reviewer_fxm7 · 2021-08-30

**Rating:** 4

**Review:**

This paper includes very short - more or less one-sentence statements in each section. The authors should aim to expand every section with greater detail so that the manuscript reads like a research paper. While this paper may simply be describing an existing baseline, there are inevitably design and training decisions that are made that will impact performance, and so this work still deserves to be written up like any other approach.

---

### Official Review · Reviewer_NLHt · 2021-08-30

**Rating:** 3

**Review:**

The authors will need to consult the kits21 paper template for a list of prompts that should be addressed. The paper cannot be published in its current form and will need a substantial rewrite. This is fine for an initial intention-to-submit submission, but please rewrite and submit your revision as soon as possible so that we can get started on re-reviewing it.

---

### Decision · Program_Chairs · 2021-08-30

**Decision:**

Major Revisions

**Comment:**

Please address the reviewer comments and resubmit